# Potential of Technosols Created with Urban By-Products for Rooftop Edible Production

**DOI:** 10.3390/ijerph17093210

**Published:** 2020-05-05

**Authors:** Baptiste J-P. Grard, Nastaran Manouchehri, Christine Aubry, Nathalie Frascaria-Lacoste, Claire Chenu

**Affiliations:** 1UMR ECOSYS, INRAE-AgroParisTech, Université Paris-Saclay, 78-850 Thiverval-Grignon, France; claire.chenu@agroparistech.fr; 2UMR SayFood, INRAE-AgroParisTech, Université Paris-Saclay, 75005 Paris, France; nastaran.manouchehri@agroparistech.fr; 3UMR SAD-APT, INRAE-AgroParisTech, Université Paris-Saclay, 75231 Paris, France; christine.aubry@agroparistech.fr; 4Ecologie Systématique Evolution, CNRS-AgroParisTech, Université Paris-Saclay, 91400 Orsay, France; nathalie.frascaria-lacoste@agroparistech.fr

**Keywords:** technosol, urban agriculture, rooftop farming, urban waste, urban farming and green roof

## Abstract

Urban agriculture is sprouting throughout the world nowadays. New forms of urban agriculture are observed such as rooftop farming. In the case of low-tech rooftop farming projects, based on recycled urban waste, one of the key issues is the type of substrate used, as it determines the functions and ecosystem services delivered by the green roof. Using a five year experimental trial, we quantified the food production potential of Technosols created only with urban wastes (green waste compost, crushed wood, spent mushroom), as well as the soil fertility and the potential contamination of food products. Regarding food production, our cropping system showed promising results across the five years, in relation with the high fertility of the Technosols. This fertility was maintained, as well as the nutrients stocks after five cropping years. Most of the edible crops had trace metals contents below existing norms for toxic trace metals with nevertheless a concern regarding certain some trace metals such as Zn and Cu. There was no trace metal accumulation in the Technosols over time except for Zn. This study confirmed that constructing Technosols only from urban wastes is a suitable and efficient solution to design rooftops for edible production.

## 1. Introduction

In the last decade, urban agriculture has been a growing topic for urban stakeholder’s worldwide. Urban agriculture is perceived as one way to counter some of the negative impacts of urban growth and development. Indeed, urban agriculture has potentially important and diverse functions, such as improving social cohesion [1], food production [2,3,4], urban waste valorization [3], improved nutrient cycling etc. The multi-functionality of urban agriculture could help to tackle several societal challenges of urbanization [5].

Whether urban agriculture can significantly contribute to feeding urban citizens has raised debate and promoted a number of empirical studies. Weidner et al. [6] showed that a wide range of methodologies are used to estimate productive surface areas, yields and other production parameters in urban agriculture, which results in a great variability of estimates. From 1.87% to 150% of vegetables and fruit demands could be met by urban agriculture in different cities depending on the type of space considered as being available for urban agriculture, the growing system used (low and/or high tech system) and on the type of production (vegetables and/or fruits and/or other multiple food categories) [6]. MacRae et al. [4] estimated that cropping all the potential growing space of the city of Toronto, including rooftop farming, would provide for 10% of the city needs in fresh vegetables, but stressed the need for more agronomic knowledge regarding this type of production. It is clear that current published estimated are more based on estimated and extrapolated yields rather than based on field/case studies [6]. 

At the ground level, urban agriculture projects are often limited by the access to land and by soil contamination. To face that, one solution is to use vertical space, e.g., the Z-Farming project (zero-acreage farming, see [7,8]). Rooftop food production also has inherent constraints, when compared to ground level food production, such as the need to limit the weight on the buildings, the need to create soils and the meteorological condition. Meteorological conditions on rooftops can be harsher compared to ground level with potentially higher access to light and more windy conditions. Depending on the context this can represent an opportunity (light) or a constraint (wind). However, little is currently known about how the constructed soil characteristics or Technosols characteristics for rooftop gardening determine the food production potential.

Only a few studies have investigated the productivity of rooftop (~12 studies, see Table 1). Whittinghill et al. [9] showed the feasibility of food production on a fertilized and irrigated extensive green roof with 10.5 cm thickness of substrate. They report a satisfying level of production except for pepper, and do not find much difference between yields at the ground and roof level. Contrarily, [2] showed a greater food production on rooftop than at ground level. Orsini et al. [3], in Bologna (Italy) demonstrated a higher production on a fertilized substrate than on a hydroponic system with simplified management. These studies highlighted that producing food on rooftop is a challenge and raised research questions regarding soil fertility, contamination of the food products as well as regarding the parent materials used to design a productive substrate on rooftops. 

Rooftop agriculture implies creating soils on the roof. The growing medium used can be called in this case isolatic Technosol [10] as it aims to stay for several years. Indeed, soil creation implies arranging and mixing different parent materials in order to obtain a functional isolatic Technosol. This functionality is directly linked to the expected ecosystem services (food production, water retention, carbon storage etc.) as well as the creation of an ecosystem design by human action and relying on ecological and biological process (biodegradation, mineralization, lixiviation, nutrient uptake etc.). Human action, through the choice of parental materials, plants, technical operations etc., should aim to maximize the functionality of such an ecosystem regarding its specific constraints. For weight constraints, natural soils are often avoided. Several experiments used peat or potting soil, which can be seen as non- or little renewable resources, while urban wastes represent a large and not sufficiently used resource. Life cycle assessment (LCA) analysis showed that a classical substrate based on peat generates a higher environmental impact than growing media based on urban wastes [11]. However, only a few studies aimed to design Technosols for productive rooftops based mostly on by-products derived from organic urban wastes, which have the advantage of being a local renewable resource and of being light (Table 1; [2,12,13]. Eksi et al. [2] used green waste compost mixed with expanded clay pellets, finding an optimal ratio of around 60%–80% of green waste in the Technosol. Aloisio et al. [14] compared on a rooftop of New-York (USA) two commercial mixes for extensive green roofs to a potting soil and measured a higher production with the potting soil over 45 days. These studies show the key influence of the nature of the parent material that directly impacts soil fertility and in consequence food production. However, among the 10 studies published on productive green roofs (Table 1), none of them looked at the spatial arrangement of the used parent materials and the study time was in average of 16 months, while such set-ups are expected to be implemented for several years. Technosols made of organic materials that are decomposable can be hypothesized to change with time, e.g., decreased particle size and porosity, changes in the organic C and N content [15,16], raising the question of the sustainability of urban agriculture based on the use of by-products generated from organic wastes from the city. Compared to classical green roof substrates such as peat or pozzolan, the used of by-products derived from urban waste as Technosol parent materials could result on more intense biological and ecological processes because of biodegradation.

Urban food production could be contaminated via two major pathways: root plant uptake and/or atmospheric fallout. Contamination of food products with trace metals is of particular concern in urban agriculture [23,24,25,26,27,28,29]. These studies show the risk of vegetable contamination due to historical pollution of urban soils and agricultural practices (such as the use of contaminated wastes). Several factors have been reported to influencing food products contamination: climate, soil characteristics (pH, organic matter etc.) [30], type of pollutant, type of plant and soil organisms [31]. An effect of urban pollution is described for different forms of urban agriculture using uncontaminated soils at natural levels trace metals concentrations [32], hydroponic system on rooftop [33] or urban contaminated soils [34,35]. However, to our knowledge the quality of vegetables grown on Technosols made with by-products derived from urban wastes was not informed.

The use of by-products derived from urban wastes, in particular of organic urban wastes, to construct Technosols appears as an opportunity for the development of rooftop agriculture, but requires an extensive evaluation. Here, we aimed to test constructed Technosols made of different by-products derived from organic urban wastes and with different layouts, in terms of (i) their fertility and (ii) fertility sustainability with time and (iii) the associated risks of food contamination with trace metals. For this, we monitored Technosols characteristics, yields and food products trace metals concentrations for Technosols made of various parent materials and with two layouts, either as layers or homogeneous, in a rooftop experiment over five years, i.e., a much longer time period than considered previously in the literature. 

## 2. Materials and Methods 

### 2.1. The Research Project T4P 

Since 2012, the research project T4P (http://www2.agroparistech.fr/T4P-un-Projet-de-recherche-innovant-pour-des-Toits-Parisiens-Productifs.html) (Pilot Project of Parisian Productive Rooftops) takes place on the rooftop of AgroParisTech (French Technical University of Agronomy; 16 rue Claude Bernard, Paris 75 005; coordinates: 48°50′24.4” N, 2°20′54.5” E). Different experimental trials address the potential for food production, ecosystem services delivery and the temporal evolution of the Technosols. The experimental roof is under a typical Marine West Coast Climate (Cfb—Köpen climate classification) in a housing neighborhood where this roof is the highest in a perimeter of 200 m.

Wooden containers, classically used as a backyard composters, are used as experimental units. These small spatial units are easy to manage and limit the weight on the roof (Figure 1).

#### 2.1.1. Experimental Trials

(1)Constructed Technosols

We used three types of urban organic by-products as parent materials of the Technosols: Green waste compost, from urban public parks and green spaces. Our supplier was a company located in Versailles near Paris, “BioYvelinesServices”.Crushed wood, made of pruning waste from urban tree crushed coarsely. The supplier was also “BioYvelinesServices”;Spent mushroom substrate, based on coffee grounds. This substrate is used to produce the mushroom *Pleurotus Ostreatus* by an urban company: “La boîte à champignons”. This firm, in partnership with a leading coffee supplier, collects used coffee grounds through a specific supply chain from coffee machines in Paris.

Woodchips (0–40 mm) were also used each as mulch in each box (a layer of 3 cm), in order to minimize substrate evaporation and weeds. Woodchips also came from the firm “BioYvelinesServices”. 

Each plot was organized identically (see Figure 2): it was filled at the bottom with 5 cm of expanded clay pebbles used as a water reserve and surrounded by an EPDM (ethylene–propylene–diene monomer) membrane. On the top of this we placed 30 cm of growing substrate surrounded by a “geotextile” through which the roots could grow.

(2)Experimental Design

Our cultivation system was inspired by an original gardening practice born in the United States of America and popularized in France a decade ago: lasagna beds [36]. The idea is to mimic a natural soil by putting down layers of “brown” and “green” organic matter. The “color” relates to the decomposability and mineralization rate of the material. For instance, a brown layer can be made of crushed wood having a low rate of mineralization and providing an input of carbon, with a high C/N ratio. The green layer can be a green waste compost with a higher mineralization rate and an input of nutrients due to a lower C/N ratio. In our case, every year, at the beginning of the cropping season, we added an additional layer of “green” matter (see details below). This ensures an input of organic matter providing nutrients by mineralization to compensate for that used by the previous crop and ensures a sufficient volume for root anchorage. 

(3)Technosols Design

Our experimental system consisted of 12 wooden boxes of 0.64 m^2^ (80 × 80 cm) each, with 0.5 m between them. In order to test the effects of Technosols nature and organization, three different types were set up in March 2012 and one in March 2015, each with three replicates (Figure 3):Lasagna* (L): a 15 cm layer of green waste compost which covered a 15 cm layer of crushed wood.Lasagna non initially inoculated* with Residues of spent mushroom substrate (L-R): a 12.5 cm layer of green waste compost covered, over a 5 cm layer of spent mushroom substrate and a 12.5 cm layer of crushed wood.A non-initially inoculated* mix (M): 30 cm of green waste compost and crushed wood mixture (50/50 *v/v*).Lasagna* (L_Bis): a treatment that had the same composition as L but was installed in March 2015 (3 years later).

The two-layers lasagna (L and L_Bis) were initially inoculated with adult earthworms belonging to three species and two ecological categories: 15 *Dendrobaena Veneta* individuals (epigeic earthworm), 35 *Eisenia Fetida* (epigeic earthworm), and 10 *Lumbricus Terrestris* (epi-anecic earthworm). Inoculation densities were lower than usual practice according to Pey et al. [37]. However, after one year, only *Eisenia Fetida* had survived with a high rate of reproduction. After two years, these earthworms had colonized every treatment, hence the different Technosols cannot be distinguished from their earthworm communities. 

In the remainder of the article the initial materials, i.e., green waste compost, crushed wood and spent mushroom substrate, will be called “parent material”. 

#### 2.1.2. Crop Rotation and Cropping Practices

(1)Crop Rotation

The crop rotation for the first two years was the same with a sequence of lettuce (*Lactuca sativa*) then cherry tomatoes (*Lycopersicum esculentum* var. chery) and green manures (*Trifolium incarnatum* and *Secale* cereal) designed to represent the most common crops grown in vegetable gardens in Paris [38] as well as to vary by their nutrients needs over the cropping season. Indeed, the tomatoes plants export more nutrients than the lettuce [39]. After two years, the rotation was diversified with four main principles:Maximize the use of space (vertically and horizontally);Associate crops whenever possible;Grow vegetables that are accurate in small growing space;Try to avoid the same family of vegetables in four succeeding years.

As shown in Table 2 these principles were not always strictly respected but did define the most important point of the rotation. At harvest, roots were returned to the soil. The only exception was for green manure left to decompose on the surface of the soil. Most of the crop grown were transplants except for beans, onions in 2016, garlic, mesclun and radish that were sown. 

(2)Pest Management

In terms of crop protection, we applied copper sulphate only on the tomato plants, in three treatments in 2012 (June, July and August) and two in 2013 (June and July). In parallel we did two preventive horsetail treatments in 2013 on tomatoes and in 2013 we put one Indian carnation plant in each box to avoid aphids. During the following years, we only used episodically some black soap and milk to avoid aphids and prevents fungi especially on zucchini.

(3)Irrigation

The experimental plots were irrigated both with rainwater and tap water. Tap water was distributed thanks to a drip irrigation system to minimize the loss due to evaporation. Since 2014 a sprinkling irrigation was installed and used only for seed germination. Water meters were installed in March 2013 to record water consumption. As no humidity sensors were installed to control this parameter, water consumption varied between the different years due to climatic condition, nature of the crop and subjective decision regarding the irrigation needs.

(4)Farm Inputs

In the design of our experimental system, we decided not to use any chemical fertilizer. The only input allowed was organic and was applied at the beginning of each cropping season (end of March or April) when the containers were re-filled with green waste compost (the lasagna systems) or spent mushroom and green waste compost (L-R) or a 50% green waste and 50% crushed wood mixture (M treatement) to their initial height of 30 cm, to compensate for substrate compaction and biodegradation.

(5)Meteorological Conditions

Temperature, wind speed, solar radiation, and rainfall were recorded at the nearby meteorological station directly on the rooftop of AgroParisTech (Campbell—CR1000) or using the “Meteo France” network with a station situated at Montsouris Park in Paris (48°49′18” N and 2°20′16” E), approximately 2.5 km from the experimental roof.

### 2.2. Data Collection

#### 2.2.1. Food Production (Quantity and Quality)

Each element of edible biomass was harvested and weighed to measure yields. Yields were measured at the plot level (0.64 m^2^) and expressed in the results section (see below) per 1 m^2^ using a linear extrapolation of the plot surface area. For measuring quality criteria of edible biomass, at least 100 g of fresh matter were collected from each container (triplicate sampling). To comply with good consumer practices, all samples were washed. The washed samples were dried at 40°C. The fresh and dry matters were weighted in order to estimate the water content in vegetables. They were then mineralized using a digestion block (DigiPREP SCP SCIENCE) and aqua regia procedure according to NF X 31,415 using hydrochloric and nitric acid. Five trace metals, cadmium (Cd), lead (Pb), copper (Cu), zinc (Zn) and mercury (Hg), currently found in polluted urban garden soils were analyzed. Atomic absorption spectrometry (AAS), using multi-element German instrumentation (Analytik Jena contra 800D, 2006) and polarized Zeeman atomic absorption spectrophotometer model Z5000 (HITACHI), was employed. Furnace atomization mode was used for Pb, Cu and Cd absorbance measurements. Flame atomic absorption spectrometry and cold vapor atomic absorption spectrometry were used to determine Zn and Hg, respectively. All measurements were performed in triplicates.

Measured contents are compared with reference values (regulatory values or mean values observed in French food). European regulation (Commission Regulation E.C., No 1881/2006) (https://eur-lex.europa.eu/legal-content/FR/TXT/?uri=CELEX:02006R1881-20180319) fix threshold values for only Pb and Cd in fresh fruits and vegetables. These values are different for different categories of vegetables (fruit, root or leaf). For Hg, the French recommended maximal concentration of 0.03 mg/kg fresh weight) is used in this study [40]. Zn and Cu being essential oligo-elements, their contents in vegetables are not regulated. Reference values used for this study correspond to 10 mg·kg^−1^ of dry matter for Cu [41] and 95 mg·kg^−1^ of dry matter for Zn [42].

#### 2.2.2. Technosol Thickness

Height was recorded at the beginning of each growing season using a simple system of level difference between the upper and the lower point of the box and the Technosol surface. Ten measurements were recorded along the diagonal of each plot using a graduated stick.

#### 2.2.3. Technosol Characteristics

Parent materials were analyzed at different times as well as the Technosols. The upper (10/15 cm) and lower layer (10/15 cm) of Technosols were sampled after two and five years using soil core samplers. A quantity of 500 g of soil samples were used for agronomic and pollutant analyses and were dried at 40 °C and crushed to pass at 2 mm sieve prior to analysis. The following methods were used by the soil laboratory of INRA Arras for pH-water [ratio soil/solution = 1/5 *v/v*; NF ISO 10390], organic carbon content (dry combustion by heating at 1000 °C with O_2_—[NF ISO 10694]), total nitrogen [NF ISO 13878]. Bulk density was measured according to NF EN 13041.

Five trace metal elements (Cd, Pb, Cu Zn and Hg), currently found in polluted urban garden soils were analyzed in the parent materials and Technosols. The samples were digested by the aqua regia method according to NF X 31,415 to measure pseudo-total concentrations of trace metals in substrates. Atomic absorption spectrometry (AAS), using polarized Zeeman atomic absorption spectrophotometer model Z5000 (HITACHI), was employed. Furnace atomization mode was used for Pb, Cu and Cd absorbance measurements. Flame atomic absorption spectrometry and cold vapor atomic absorption spectrometry were used to determine Zn and Hg, respectively. All measurements were performed in triplicates.

### 2.3. Statistical Analysis

Statistical analyses were performed using R software (R-3.1.1). The three treatments with three replicates for each case were compared using an analysis of variance after ensuring the normality of the data using a Shapiro test. A multiple comparison of means was determined by the post-hoc Tukey test. When normality of data was not respected, a Kruskal–Wallis test was applied followed by a post-hoc Nemenyi test. A significance level of *p*-value < 0.05 was used for each test.

## 3. Results

### 3.1. Food Production and Soil Fertility

#### 3.1.1. Yields

Table 3 shows the yield evolution during five growing seasons per crop and treatment. Comparison should only be achieved for a specific crop the same year. Exceptions were the first and second growing seasons that could be compared, as the same crop were grown. There, the second year showed higher yields than the first one for L and M (mainly due to cherry tomatoes) and a steady yield for L-R. There was no decrease of average yields with time. Average yields for all treatments ranged from 5.8 kg·m^−2^ the first year to 19.6 kg·m^−2^ for the fifth growing season. Year, treatment and crop showed a significant effect on yields. The most productive treatment was L-R with 77 kg·m^−2^ produced during the five years, i.e., an average yield of 15.4 kg·m^−2^.year^−1^ compared to 62.1 kg·m^−2^ (i.e., 12.4 kg·m^−2^·year^−1^) for L and 41.7 kg·m^−2^ (i.e., 8.3 kg·m^−2^·year^−1^) for M. Regarding professional yields, fruits or roots vegetables seemed to better perform on constructed Technosols with equivalent or even higher yield than leafy vegetables. Indeed, lettuce presented lower yields than professional ones in any tested situation.

#### 3.1.2. Factors Impacting Food Production

The results presented in Table 3 show a clear effect of the treatment depending on the type of crop. Based on the comparison of the four treatments, three effects could be observed:Regarding the impact of the spatial arrangement of the parent material (comparing L to M): L showed a higher productivity than M over the 5 years (Table 3). However, this effect was mainly and almost only attributed to leafy vegetables. No significant differences could be noted for the four fruits vegetables grown and only one significant difference for the root vegetables.Ageing of Technosol little impacted yields, as over the seven crops growing on either young (L_bis) or aged (L) Technosol (Table 3), only yields of carrots were higher on the young Technosol.

Nature of parental material only weakly impacted yields (comparison of L to L-R). Only two crops among 16 showed a significant yield difference where L < L-R (Table 3). Nevertheless, in terms of total and average yield over the five growing seasons, L-R showed significantly higher yields than L (see Table 3).

#### 3.1.3. Technosol Evolution and Fertility

The thickness of the Technosols decreased strongly in the first year, with an average subsidence, i.e., a gradual lowering of its surface elevation, of 39% for all treatments (Figure 4), and thereafter the thickness of the Technosols remained between 8% to 21% of the initial height. Subsidence was the same for all treatments. The L_Bis treatment showed the same pattern as L with a sharp subsidence the first year (33%) and less the second year (22%; Figure 4).

Bulk density of the Technosols after five years was higher than that of the parent materials, showing that the observed subsidence was associated with Technosol compaction. It was the same for both layers after five years, but was significantly lower in the L-M Technosol, compared to the layered ones (Table 4). The increase in bulk density resulted in a slight decrease of total porosity that remained between 30% to 40% of the total volume (Table 4).

Parent materials presented contrasted characteristics regarding different parameters (see Table 4 regarding organic content, CEC, C/N, Ntot. etc.). Chemical characteristics of the different treatment showed a decrease of the carbon content of the different layers (Table 4) of 0%–20% for the upper layers made initially of compost, and yearly replenished with compost (L and L-R), of 19% to 30% of the upper and lower layers made initially of a 50% mixture of compost and crushed wood (M) and of 37% to 53% of the lower layers made initially from crushed wood (L and L-R). For all layers and treatments, the organic C content decreased in average by 14% between year two and five. The total nitrogen content increased markedly compared to the parent material, in particular for the lower layers of L, L-R and L_Bis where it increased by a factor of two, and accordingly the C/N ratio decreased. A similar evolution was noticed when comparing N contents and C/N ratio between year two and five. The CEC always increased, compared to parent materials and over time, and was high in all Technosols and layers (average of 56.5 ± 4 cmol^+^.kg^−1^) (Table 4). Regarding nutrients, nitrate contents generally decreased compared to parent materials while ammonium contents increased (Table 4). However, in the lower layers made of crushed wood, total nitrogen content generally increased compared to parent materials. Regarding P, organic P tended to increase compared to parent materials and over time, while there was no unique trend regarding P_Olsen_. Lastly, available K, i.e., exchangeable K, decreased compared to parent material and over time.

Over five years, total constructed Technosol dry weight increased in average by 37% for all treatments with a larger weight increase for treatment with layers (52% of increase over five years) compared to M treatment (7% increase) (see Table 5). The calculation of element stocks in the Technosol (Table 5) was consistent with the evolution of bulk density and concentrations (Table 4): over time Corg stocks steadily decreased (except for L) while CEC and total N stocks increased.

Regarding the design of constructed Technosols, sharp differences appears while comparing L and M. While the total dry weight of the Technosol increased by 6% for M in 5 years, a 10-fold increase was observed for L, within that case 72% of the dry weight gain for L due to the lower layer (Table 5). The same pattern could be observed for Corg, CEC and N total with, in each case, differences between lower and upper layers. L and L-R presented common features, except for the evolution of Corg in the lower layer, which was stronger for L (Table 5). Finally, it is interesting to notice that after 2 years and with a common pattern regarding thickness evolution (Figure 4), weight gain in the upper layer of L_Bis was equivalent to L, while the mass gain in the lower layer represented half of the weight gain of L after five years. Except for the stock of Corg in the lower layer, the evolution between L and L_Bis regarding Ntot and CEC are equivalent with a higher increase of Ntot in the lower layer of L after 5 years.

### 3.2. Harvest Contamination

#### 3.2.1. Trace Metal Contents in Vegetables

Pb, Cd, Hg, Cu and Zn were measured in nine species of fruit-, root- and leafy-vegetables during the five years of experimentation (Figure 3). Except for copper, fruit vegetables showed the lowest trace metals concentrations (see Figure 5). Leafy vegetables presented generally lower values than roots vegetables except for zinc. Regarding Cd, all products presented concentrations below the European threshold values, ranging between 0.7% and 23% of the norm with an average of 5% of the norm values. For Pb, all vegetables had concentrations below the threshold values (ranging from 3% to 76% of the norm values with an average of 29% of the norm) except for seven onion samples, presumably due to the presence of substrate particles trapped in these samples despite the respect of the good practice of consumption by washing all samples. Finally, Hg contents were below the threshold value for all vegetables ranging between 1.29% and 21.9% of the recommended reference value. Contrary to the potential toxic trace metals, Cu et Zn contents exceeded the reference values in some vegetables ranging from 23% to 318% of the reference value with an average of 123% for Cu and ranging between 22% to 226% of the reference value for Zn with an average of 79%.

#### 3.2.2. Trace metals in Technosols

Pb, Cd, Hg, Cu and Zn contents in Technosols and parental materials (Table 6) did not exceed the French reference thresholds corresponding to growing substrates (NFU 44-551) nor do those of soils receiving sludge (national decree n 97-1133, 08/01/1998 (https://www.legifrance.gouv.fr/affichTexte.do?cidTexte=JORFTEXT000000570287). Cd and Hg values were below regional thresholds corresponding to pedogeochemical background levels [44], while Zn and Cu contents exceeded the regional threshold values by a factor of 3 and 2, respectively.

For upper and lower layers, the only significant variation of content was observed for Zn with a raised in average of 15% compared to the initial content of green waste compost. Comparing the value of parent material to each layer, showed a significant increase for all trace metals regarding the lower layer. No significant differences were observed between treatments.

## 4. Discussion

### 4.1. Productivity

Expected productivity and self-sufficiency from urban agriculture has started to be debated in the literature [6]. The lack of actual data from urban agriculture weakens this debate and hinders the up-scaling of urban agriculture. In the present study we measured relatively high productivity per square meter (11.5 ± 5.3 kg·m^−2^ in average for all treatments during 5 years), compared to average yields in community gardening (1.2–2.6 kg·m^−2^) [6,39], in horticulture in developed countries (2.5–3.3 kg·m^−2^) and in professional and intensive gardening (5.4–7.1 kg·m^−2^; [6]. Our measured yields were close to those of other rooftop studies based on Technosols made out of organic materials [3], which confirms the interest of using urban wastes as already concluded in other articles [4,21,22]. Nevertheless, caution should be taken when comparing yield of different vegetables and in different area of the world due to climate differences, different water content of vegetables, different vegetables varieties as well as differences in the surface area considered. In rooftop farming, a minimum surface area needs to be dedicated to other uses than food production, e.g., pathways. Here, we expressed yields per productive surface area, while Orsini et al. [3] consider that only 65% of the roof surface could be dedicated to food production. In our cases, if we take this percentage of 65%, the average yield of L-R would, during the five growing seasons, decrease from 14.7 kg·m^−2^ to 9.6 kg·m^−2^ at roof level. Caution should then be taken to clearly define the surface considered. In addition, other criteria could be taken into account when comparing food production by different urban agriculture systems, such as the size of the vegetables, marketable yield, nutritional quality etc. [9]. Such parameters should be considered in future studies on urban agriculture.

### 4.2. Food Quality

In terms of vegetables quality, the results confirmed generally the well-known characteristic of plant-specific contamination [41] dealing with higher levels of trace metals in root and leafy vegetables compared to fruit vegetables. The obtained results are in agreement with other studies showing a low contamination of produces growing on organic substrates with natural trace metal occurring [32]. Here, the Technosols were relatively basic (pH > 7) and rich in organic matter limiting thereby the trace metals transfer towards the crops. However, organic matter contents could inhibit or promote the trace metal soil/plant transfer [45], depending on the trace metal affinity for organic matter. Still, seven onion samples were found to be above limits, likely due to the potential presence of soil particles trapped in samples despite the washing practice. High levels of Zn and Cu in vegetables could also be related to high concentration of these two trace metals in the parent materials, exceeding the background values of Zn and Cu in soils of the region [44]. Zn is considered as a mobile trace metal and could be easily desorbed from the soil solid phase. Cu has high affinity for organic matter and could be remobilized as organic soluble complex. Cu and Zn are presents in atmospheric particulate matters in urban zones [46]. They are known for their capacity of foliar absorption in leafy vegetables [42]. However, Cu and Zn are not regulated as toxic element for human and their hazard is mostly associated to their deficiency [41].

The experimental plan was not designed to investigate the dominant contamination pathway and trace metal found in vegetables could originate from soil and/or from atmospheric wet or dry fallout: trace metals contents in vegetables involve thus all potential exposure pathways. For example, Cu and Zn high contents in some vegetables could be also due to particles present in the urban atmosphere. The trace metal content of Technosols did not significantly increase during the 5 years of experimentation (Table 6), except for Zinc which increased by 15% of its initial value and may be explained by atmospheric fallout.

### 4.3. Technosol Fertility

Factors influencing food production on rooftops has been little studied so far. Here, we were able to compare Technosols made from different parent materials and with different layouts. Overall, the Technosols were very fertile, exhibiting a large porosity, much larger than in mineral soils used for market gardening, with a neutral to slightly alkaline pH. Available contents of P and K (Table 4) were much larger than the P and K contents above which DEFRA Fertilizer manual [47] recommends not to fertilize vegetable crops: P index 4 (46 to 70 mg Olsen P. L^−1^ soil which corresponds to 0.2 to 0.25 gP·kg^−1^ Technosol with a bulk density of 0.35 g·cm^−3^) and K index 4 (401 to 600 mg exchangeable K. L^−1^ soil, which corresponds to 0.67 to 0.82 cmol exchangeable K·kg^−1^ Technosol). Regarding nitrogen, the contents of both total N and mineral N were very large in the Technosols compared to mineral soils of urban vegetable gardens [48].

Regarding chemical fertility, stocks of total N and P increased over time, presumably as the consequence of the yearly organic matter inputs on the one hand and the accumulation of N and P released by mineralization of the Technosols organic matter on the other hand. However, the amount of total P and N lost by leaching should be measured in future studies. Both available P (Olsen P) and K (exchangeable K) decreased over time, but as mentioned previously, they remained well above standard values for no fertilization recommendations, showing that the fertility of the Technosols is not at stake over five years in the present systems.

### 4.4. Technosol Design and Evolution

While the nature of the parent materials had little effect on yields, layered Technosols allowed for significantly higher yields than homogeneous ones, in which compost and crushed wood had been mixed (Table 3). Our results do not allow to explain clearly this effect, but we propose two hypothesis that are not exclusive: (i) a lack of nitrogen when compost and crushed wood are mixed; (ii) different availabilities of air and water in the Technosol due to the initial layering of parent material that evolve across time. In fact, regarding the data, much of the yield differences between L and M occurred during the first three years with on leafy vegetables.

Regarding the thickness evolution and observed subsidence, we could distinguish two phases: a first phase with a strong subsidence during the first growing season and a second phase after. With time and our cultural practices of refilling the upper layer, the characteristics of the two layers became closer, with large changes in the lower layers. Our results clearly showed that different pedogenic processes occurred in the upper and lower layers of the Technosols. The upper layer was likely affected mainly by biodegradation and mineralization, resulting in a decrease of its organic matter content and a drop of the C/N ratio.

The lower layers showed the strongest evolution with increased in C and N contents, CEC, nutrients over time. This can be explained by biodegradation, i.e., a fast decomposition of the crushed wood as well as by transfer of upper layers particles or solutes downwards, by gravity, by leaching and/or by the earthworm’s activity. Illuviation and/or lixiviation are hence taking place in the profile of the constructed Technosols. As for other Technosols, pedogenesis processes similar to those occurring in natural soils were observed [49], i.e., biodegradation and illuviation at however a higher rate than in natural soils. The observed subsidence may be related to that observed in peat soils upon draining that allows fast organic matter mineralization (e.g., [50]). In our Technosols, the high content of organic matter, available water (irrigation) and available oxygen (large porosity of the Technosol) may explain the high intensity of processes such as biodegradation.

This pedogenesis may compromise the sustainability of the constructed Technosols for growing vegetables if the subsidence leads to a compaction and reduction of soil porosity that compromises root growth. However, we did not observe such a porosity loss over five years (Table 4). Soil structure formation was not investigated here, but new pores and aggregates genesis may take place, in particular due to the activity of earthworms. The biodegradation may also compromise the sustainability of the fertility of the constructed Technosols, if, eventually, all labile organic components of the Technosols having been mineralized, only recalcitrant organic compounds remain and not enough mineral elements are released by mineralization to meet the plants demands. However, it is likely that the remaining organic matter would have a large CEC hence retaining nutrients, and yearly addition of new by-products from urban organic wastes adds new N, P and other nutrients to the system.

So far, no published study monitored soil characteristics and yields in rooftop agriculture for periods longer than two years (Table 1). Here we could assess the sustainability of the fertility of the Technosols by comparing their characteristics after two and five years and by measuring yields over five years. Yearly additions of compost were performed, in all wooden boxes on the top layer of the Technosol, to compensate for the observed subsidence and for the presumed loss of elements due to exports by the harvested vegetables or by leaching. Over years, yields were maintained and even increased (Table 3) demonstrating that Technosols fertility remained high enough. Regarding physical evolution, despite a strong shrinkage, the porosity remained large (Table 4). A decrease in porosity was observed in other studies and explained by the biodegradation of organic materials [15,16].

While an ageing of Technosol was observed, their fertility was maintained over five years and this roof top farming system appeared sustainable over 5 to 10 years, while the practice of adding some compost yearly is maintained. Other urban wastes such as urine and/or dry bones could also be added to fertilize the soils.

## 5. Conclusions

Developing rooftop farming requires to design specific growing systems fitted to the constraints of this environment: weight constraints, specific meteorological conditions with potential higher rates of evaporation and dealing with potentially small surface areas that force to maximize the use of space and adapt the cropping system. Using by-products from urban organic waste for rooftop farming requires to select organic wastes that are not contaminated or only contaminated at a low level, and that are biodegradable and hence can provide the necessary nutrients to the plants. A balance has to be found between fast enough decomposition of the urban by-products, to ensure provision of nutrients and slow enough decomposition, to ensure that stability of the constructed Technosol, i.e., that the physical environment of roots persist long enough, a balance that requires an appropriate selection of parent materials. In the present system, the parent materials provided ample amounts of nutrients, but it was hence necessary to add large amounts of new by-product after the first year and smaller amounts the following years, which is a constraint. It is nevertheless a way to favor circular economy while creating soils of a possibly known quality regarding soil contamination and fertility. We showed that by-products from urban organic wastes such as crushed wood, green waste composts and spent mushroom substrate were valuable substrates to build highly fertile Technosols leading to acceptable and even consequent yields per square meter with vegetables respecting the European norms for regulated trace metals (Cd and Pb). Certainly, the quality of urban vegetables remains a public health concern and needs to be further investigated for a wider range of vegetable species and also for other pollutants present in urban environment such as polycyclic aromatic hydrocarbons. Our study clearly showed the feasibility of using by-product from urban organic waste as parent materials of Technosol. Thus, questions relative to the access to these by-products in terms of volume, quality and cost needs to be investigated to ensure the feasibility of this practice in different urban context.

Regarding Technosol design, a layered layout of the Technosol, mimicking the A and B horizons of a mineral soil, was more favorable to plant growth, while the reasons behind still need to be identified. While a pedogenesis was clearly taking place in the Technosols, their fertility was maintained at sufficient level over five years to maintain the yields, demonstrating the sustainability of this rooftop organic wastes vegetable cropping system. The characteristics of the material (high content of organic matter and low density) as well as the yearly input of fresh organic material seem to promote pedogenetic processes such as biodegradation of the organic matter and subsidence, partly caused by the mineralization of the organic matter. Furthermore, the increased C content and N content of lower layers can only be explained by transport processes from the upper layers downwards. Illuviation and/or lixiviation are hence also taking place in the profile of the constructed Technosols. This pedogenesis may compromise the sustainability of the constructed Technosols for growing vegetables, if the consolidation leads to a compaction and reduction of soil porosity that compromises root growth. However, we did not observe such a porosity loss over five years (Table 4). Soil structure formation was not investigated here, but may take place, in particular due to the activity of earthworms. The biodegradation may also compromise the sustainability of the fertility of the constructed Technosols, if, eventually, all labile organic components of the Technosols having been mineralized only recalcitrant or stabilized organic compounds remain and not enough mineral elements are released by mineralization to meet the plants demands. However, it is likely that the remaining organic matter would have a large CEC, retaining nutrients and changes could be implemented in the cultural system to manage plant nutrition, such as adding urine as a fertilizer, inserting legume crops in the rotation.

In the future, to up-scale urban agriculture in dense cities, the conquest of roof space (mostly unused for the moment) is necessary. To design proper and suitable growing systems, a wider variety of urban wastes could be considered and further questions need to be addressed, such as the pest management, cost analysis and the water management. Rooftop farming could be of interest for the city of tomorrow only if it functions in symbiosis with the urban environment: minimizing the use of natural resources and maximizing the use of space available to create and circular and multifunctional ecosystem.

## Figures and Tables

**Figure 1 ijerph-17-03210-f001:**
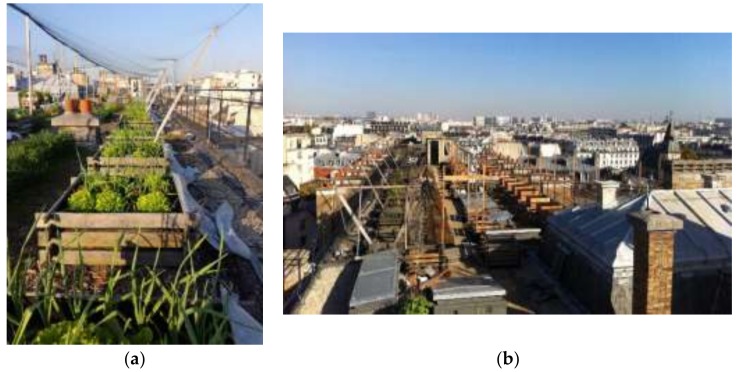
(**a**) Pictures of the T4P experimental devices in April 2017. (**b**) the experimental devices. Source B. Grard.

**Figure 2 ijerph-17-03210-f002:**
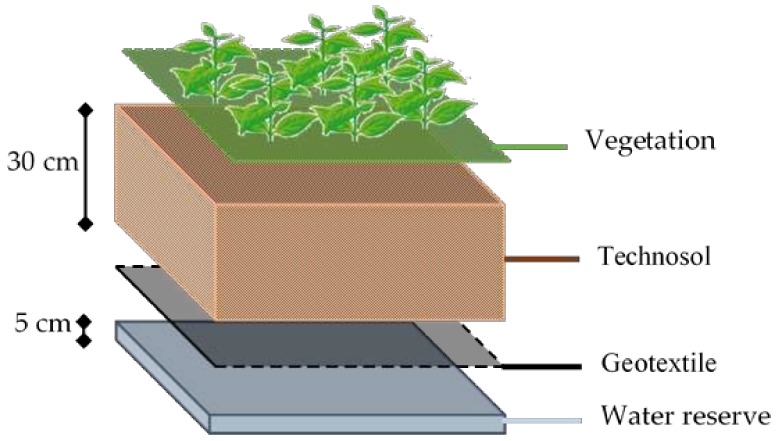
Vertical organization of an experimental plot on the T4P research project.

**Figure 3 ijerph-17-03210-f003:**
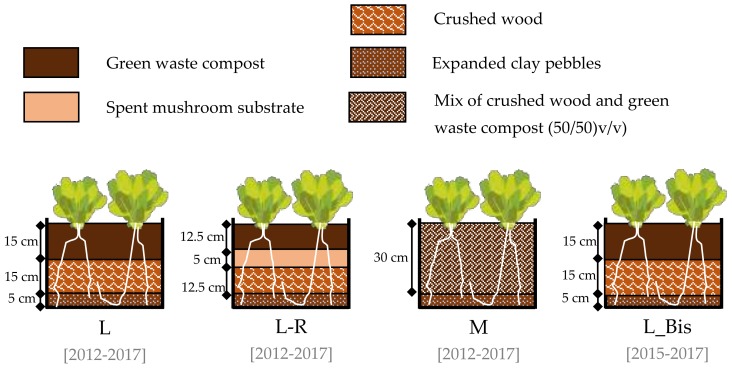
Scheme of the different experimental treatments.

**Figure 4 ijerph-17-03210-f004:**
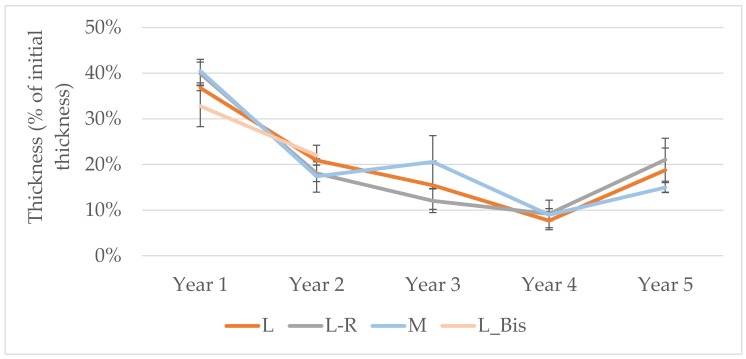
Technosol thickness evolution during the five years.

**Figure 5 ijerph-17-03210-f005:**
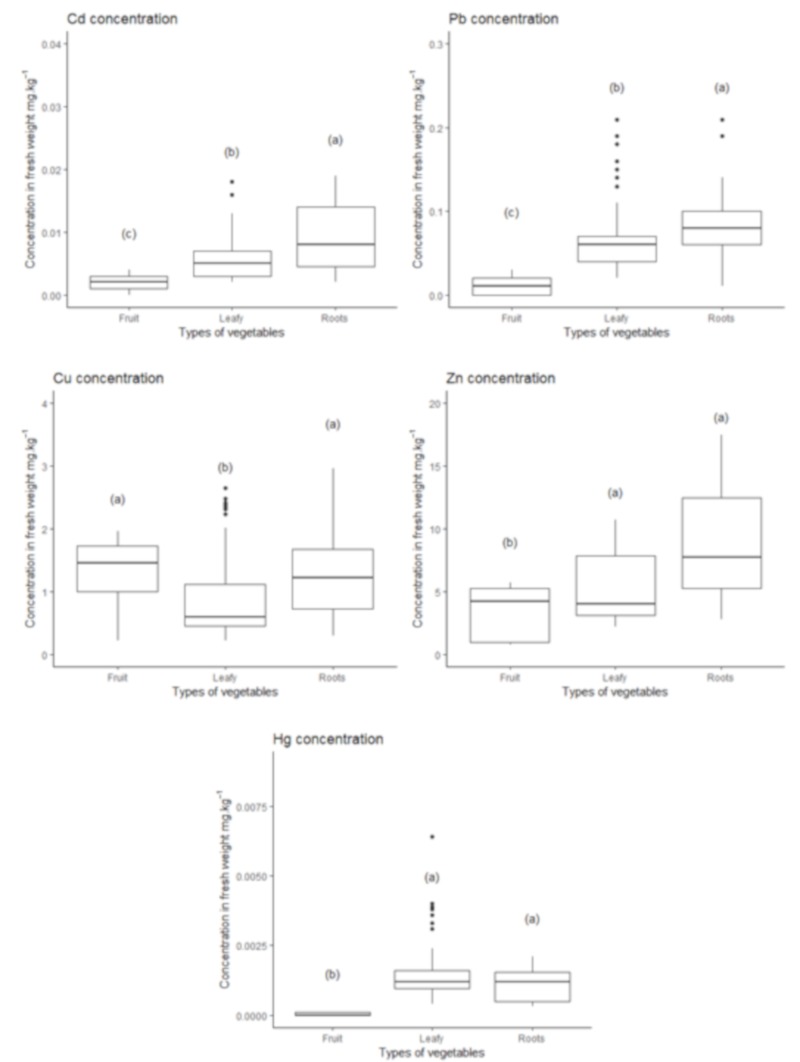
Trace metals concentrations in different types of vegetables. Leafy vegetables (3 species, *n* = 39–51 samples: lettuce, cabbage and mesclun), roots (2 species, *n* = 28 samples: carrot, onion) and fruits (2 species, *n* = 13–22 samples: cherry tomato, zucchini). (a), (b), (c), …: refers to statistical analysis. Regarding types of vegetables, two types without a common letter are significantly different (*p*-value < 0.05). Recommended maximal concentration were for Cd ⬄ 0.2 (leafy vegetables) or 0.05 (fruits vegetables) mg·kg^−1^ of fresh weight; Pb ⬄ 0.3 (leafy vegetables) or 0.1 (fruits vegetables) mg·kg^−1^ of fresh weight and Hg ⬄ 0.03 mg·kg^−1^ in fresh weight.

**Table 1 ijerph-17-03210-t001:** Case study dealing with food production on rooftop. (a): [14]; (b): [2]; (c): [17]; (d): [18]; (e): [3]; (f): [19]; (g): [9]; (h): [20], (i): [21] and (j): [22]. LCA = Life Cycle Assessment.

Ref.	Main Aim of the Study	Edible Crop Grown	Type/Composition of Substrate	Fertilization (F) and Irrigation (I)	Soil Depth (cm)	Study Time (months)	Köppen Climate Classification	Study Location
(a)	Effect of substrate and type of plant on (i) water retention, (ii) P and N loss in drainage water and (iii) food production	Amaranthus (tricolor L.; cruentus and dubius)	Extensive mix; Substrate «GaiaSoil»; Potting soil.	I	11	1.5	Cfa ⬄ Humid subtropical climate	New-York (USA)
(b)	Impact of organic matter on food production level	Cucumbers and peppers	Expanded clay pellets and sand mix with a compost from garden	F and I	12.5	6	Dfa ⬄ Humid continental climate	Michigan University (USA)
(c)	Potential for food production and water retention of substrate and hydroponic based system.	Kale	Potting soil	F and I	10.2	1	Cfa ⬄ Humid subtropical climate	Virginia Technology (USA)
(d)	Test of fertilization on productive green roof	Swiss chard	Substrate of intensive green roof (rooflite^®^): lightweight mineral aggregates, organic composted component and mushroom compost.	F and I	11	2	Cfa ⬄ Humid subtropical climate	Columbia University (USA)
(e)	Comparison of food production potential of different cropping system	Black cabbage, cantaloupe, chicory, chili pepper, eggplant, lettuce, tomato and watermelon	Topsoil and compost	F and I	20	21	Cfa ⬄ Humid subtropical climate	Bologna (Italy)
(f)	LCA of different cropping system	Same as ref. (e)	Topsoil and compost	F and I	20	21	Cfa ⬄ Humid subtropical climate	Bologna (Italy)
(g)	Evaluation of food production potential	Basil, beans, cucumbers, chives and peppers	Expanded clay pellets (50%), sand (35%) and leaf compost (15%)	F and I	10.5	36	Dfa ⬄ Humid continental climate	Michigan University (USA)
(h)	Test of different mulches and fertilization	Same as ref. (h)	Commecial substrate « XeroFlor »^®^	F and I	12.7	24	Dfa ⬄ Humid continental climate	Michigan University (USA)
(i)	Ecosystem services delivered by productive green roof	Lettuce and cherry tomatoes	Urban waste: green waste compost, spent mushroom substrate and crushed wood.	I	30	24	Cfb ⬄ Marine West Coast Climate	Paris (France)
(j)	Food production potential and contamination by productive green roof	Same as ref. (i)	Urban waste: green waste compost, spent mushroom substrate and crushed wood.	I	30	24	Cfb ⬄ Marine West Coast Climate	Paris (France)

**Table 2 ijerph-17-03210-t002:** Calendar of the crop rotation 2012–2017.

**2012**														
**J**	**F**	M		A	M	J		J	A	S	O		N	D
		*03/01 -* **Lettuce** *- 06/06*				*06/06 -* **Cherry tomatoes** *- 10/15*						*10/15 -* **Green manure**		

**2013**														
J	F	M		A	M	J		J	A	S	O		N	D
**Green manure** *- 04/02*				*04/02 -* **Lettuce** *- 05/22*		*05/22 -* **Cherry tomatoes** *- 10/14*						*10/14 -* **Green manure**		

**2014**														
J	F	M		A	M	J		J	A	S	O		N	D
**Green manure** *- 03/31*				*03/31 -***Cabbage - Kale** …										
				*03/31 -* **Beans** *- 03/10*										

**2015**														
J	F	M		A	M	J		J	A	S	O		N	D
… **Cabbage - Kale *-*** *04/02*				*04/02 -* **Zucchini** *- 09/30*							*09/30 -* **Lettuce** *- 12/07*			*12/07 -* …
				*04/02 -* **Radish** *- 06/12*										

**2016**														
J	F	M		A	M	J		J	A	S	O		N	D
… **Mesclun** *- 03/14*			*03/14 -* **Onions** *- 10/21*									*10/22 -***Onions** …		
			*03/14 -* **Carrots** *- 10/21*									*10/21 -***Lettuce**...		
												*10/22 -***Garlic** …		

**2017**														
J	F	M		A	M	J		J	A	S	O		N	D
… **Onions** *- 05/22*														
… **Lettuce** *- 05/02*														
… **Garlic** *- 05/22*														

**Table 3 ijerph-17-03210-t003:** Yield per year, per crop and per treatment. The letters correspond to statistical analysis per year allowing to compare different treatment (*p*-value < 0.05).

		Year	Number of Cropping Days	Number of Plants per m^2^	(a) or (m) *	Yield (kg·m^−2^ of Cultivated Area **)				Professional Yield *** (kg·m^−2^)
						L	L-R	M	L_bis	
Leaf	Cabbage	2014	349	6	(a)	4.9 ± 1 (ab)	6.2 ± 0.7 (a)	2.9 ± 0.4 (bc)		7.1
	Lettuce	2012	45	8	(m)	2.7 ± 0.3 (b)	4.3 ± 0.3 (a)	0.4 ± 0.1 (c)		5.8
		2013	49	8	(m)	2.7 ± 0.1 (a)	2.2 ± 0.4 (a)	0.6 ± 0.1 (b)		
		2015	67	14	(m)	1.5 ± 0.1 (a)	1.5 ± 0.2 (a)	0.6 ± 0.2 (c)	1 ± 0.1 (b)	
		2017	193	14	(a)	2.8 ± 1.1 (a)	3.5 ± 1.7 (a)	2.5 ± 0.5 (a)	3.1 ± 0.7 (a)	
	Mesclun	2015	97	-	(m)	0.7 ± 0.2 (a)	1.1 ± 0.2 (a)	0.2 ± 0.1 (b)	0.2 ± 0 (b)	
Fruits	Zucchini	2015	180	3	(a)	9.4 ± 2.7 (a)	13.4 ± 1.4 (a)	7 ± 1.6 (a)	7.2 ± 1.2 (a)	2-3.5
	Beans	2014	168	6	(a)	7.6 ± 0.8 (a)	8.1 ± 1.8 (a)	9.1 ± 0.8 (a)		-
	Tomatoes	2012	130	6	(m)	2.1 ± 0.2 (bc)	3.9 ± 0.4 (a)	2.4 ± 0.1 (ab)		4–8
		2013	144	6	(m)	4.8 ± 0.5 (a)	5.1 ± 0.3 (a)	4.2 ± 0.4 (a)		
Roots	Carrots	2016	220	-	(a)	12.3 ± 2.5 (ab)	17.6 ± 1.1 (a)	6.7 ± 3.5 (b)	16.9 ± 1.5 (a)	3–7
	Onions	2016	220	-	(a)	1.3 ± 0.4 (a)	1 ± 0.4 (a)	0.8 ± 0.9 (a)	0.8 ± 0.1 (a)	1.5–2.5
		2017	212	-	(a)	0.7 ± 0.1 (a)	0.6 ± 0.1 (a)	0.4 ± 0.1 (a)	0.4 ± 0.1 (a)	
	Radish	2015	70	-	(a)	4.1 ± 0.1 (a)	4.7 ± 0.7 (a)	1.5 ± 0.2 (b)	1.9 ± 0.3 (b)	
	Garlic	2017	212	-	(a)	2.2 ± 0.3 (a)	1.6 ± 0.5 (a)	1.4 ± 0.4 (a)	1.6 ± 0 (a)	0.2–0.5
	Green manure	2013	168	-	(m)	0.8 ± 0.2 (a)	0.6 ± 0.1 (a)	0.2 ± 0 (b)		-
		2014	167	-	(m)	1.7 ± 0.2 (a)	1.6 ± 0.3 (a)	0.6 ± 0.3 (b)		-

*: Association (a) or monocrop (m); **: surface only dedicated to food production (value extrapolated from each plot of 0.64 m^2^). *** Reference: ITAB—biological open air vegetable production by professional producer [43]. (a), (b), (c), …: refers to statistical analysis. For a same crop, two treatments without a common letter are significantly different (*p*-value < 0.05).

**Table 4 ijerph-17-03210-t004:** Physical and chemical soil characteristics over time. All results are expressed per kg of dry matter. Statistical test allow comparison between parental material (green waste compost; spent coffee grounds and crushed wood) and a single treatment. Treatment are not here comparable. (a), (b), (c), …: refers to statistical analysis. Per treatment, two layers without a common letter are significantly different (*p*-value < 0.05). GW = green waste compost; SMS = spent mushroom substrate.

Treatment	Height	Year	Bulk Density	Total Porosity	Organic Carbon	pH	CEC	CaCO_3_ tot.	C/N	Ntot.	N-NO_3_	N-NH_4_	P_2_O_5_ total	Porga.	P.Olsen	K tot.	K Coba.
			g·cm^−3^	cm^3^·cm^−3^	g·kg^−1^		cmol^+^·kg^−1^	g·kg^−1^		g·kg^−1^	mg·kg^−1^	mg·kg^−1^	g·kg^−1^	g·kg^−1^	g·kg^−1^	g·kg^−1^	cmol^+^/kg^−1^
**GW compost**			0.2 ± 0.02 (cd)	0.4 ± 0.04 (bc)	230 ± 22.6 (c)	7.8 ± 0.1 (a)	40.6 ± 1.3 (c)	56.2 ± 6.1 (a)	18.7 ± 0.8 (bc)	12.2 ± 0.8 (d)	263.5 ± 69.9 (a)	49.8 ± 5.9 (b)	4.9 ± 0.6 (c)	1.3 ± 0.2 (a)	0.6 ± 0 (b)	12.2 ± 0.38 (a)	12.9 ± 0.9 (a)
**SMS**			0.1 ± 0.02 (d)	0.4 ± 0.02 (b)	415 ± 42.9 (a)	6.7 ± 1.5 (b)	27.1 ± 3.5 (d)	37 ± 45.4 (b)	15.9 ± 1.6 (cd)	26.1 ± 1.6 (a)	5.4 ± 3.2 (b)	513.3 ± 369.1 (a)	3.2 ± 0.5 (e)	1.4 ± 0.4 (a)	0.8 ± 0.4 (a)	4.3 ± 0.9 (e)	10.6 ± 2.4 (b)
**Crushed wood**			0.1 ± 0.04 (d)	0.7 ± 0.03 (a)	454.3 ± 5.7 (a)	7.3 ± 0.1 (ab)	23.1 ± 1 (e)	3.2 ± 1.1 (c)	96.9 ± 0.3 (a)	4.7 ± 0.3 (d)	1.7 ± 0.1 (b)	53.8 ± 4.3 (b)	1.5 ± 0.14 (f)	0.6 ± 0.1 (b)	0.4 ± 0.1 (c)	6.2 ± 0.15 (d)	13.6 ± 0.3 (a)
**L**	**Upper**	2nd			234.7 ± 16.3 (c)	7.9 ± 0.1 (a)	53.4 ± 2.6 (b)	51.7 ± 2.1 (a)	16.6 ± 0.8 (b)	14.1 ± 0.8 (c)	6.7 ± 2.7 (b)	50 ± 12.7 (b)	4.9 ± 0.16 (c)	1 ± 0.2 (ab)	0.6 ± 0.03 (b)	8.8 ± 0.31 (b)	6 ± 0.1 (c)
		5th	0.4 ± 0.01 (a)	0.3 ± 0.04 (ef)	184.3 ± 21.5 (d)	7.6 ± 0.1 (a)	54.2 ± 2.7 (b)	56.7 ± 2.8 (a)	13.5 ± 1.5 (d)	13.7 ± 1.5 (cd)	15.4 ± 5.6 (b)	61.5 ± 1.1 (b)	5.4 ± 0.21 (b)	1.3 ± 0.1 (a)	0.5 ± 0.01 (c)		0.6 ± 0.4 (d)
	**Lower**	2nd			248.3 ± 22.3 (cd)	7.8 ± 0.1 (a)	53.1 ± 1.3 (b)	42.9 ± 1.8 (ab)	18.4 ± 0.6 (b)	13.5 ± 0.6 (cd)	5.8 ± 2.6 (b)	44.6 ± 10.8 (b)	4.6 ± 0.08 (c)	1.1 ± 0.2 (a)	0.5 ± 0.04 (b)	8.7 ± 0.26 (b)	7 ± 0.4 (c)
		5th	0.4 ± 0.04 (a)	0.3 ± 0.02 (f)	213.7 ± 24 (cd)	7.5 ± 0.05 (a)	58.5 ± 0.1 (a)	47.2 ± 7.7 (ab)	14.5 ± 1.3 (d)	14.8 ± 1.3 (c)	10.9 ± 1.4 (b)	62.4 ± 4.4 (b)	5.2 ± 0.28 (bc)	1.3 ± 0.1 (a)	0.4 ± 0.06 (c)		0.5 ± 0.2 (d)
**L-R**	**Upper**	2nd			242.3 ± 27.1 (c)	7.9 ± 0.1 (a)	54.6 ± 1.1 (b)	57.8 ± 1.1 (a)	14.8 ± 1.7 (d)	16.4 ± 1.7 (c)	7.4 ± 1.8 (b)	69.7 ± 1.2 (b)	4.8 ± 0.08 (c)	1.5 ± 0.7 (a)	0.5 ± 0.04 (bc)	8 ± 0.39 (b)	4 ± 0.8 (d)
		5th	0.3 ± 0.1 (ab)	0.3 ± 0.06 (def)	200.3 ± 17.3 (d)	7.7 ± 0.1 (a)	59.5 ± 2.1 (a)	56.7 ± 6.1 (a)	12.9 ± 1.6 (d)	15.5 ± 1.6 (c)	25.8 ± 10.4 (b)	63.5 ± 1.9 (b)	5.5 ± 0.25 (b)	1.2 ± 0.3 (a)	0.5 ± 0.03 (c)		0.4 ± 0.2 (e)
	**Lower**	2nd			288 ± 51.5 (b)	7.7 ± 0.02 (a)	55.7 ± 2 (b)	46.8 ± 11 (ab)	18.4 ± 0.9 (c)	15.6 ± 0.9 (c)	6.9 ± 1.3 (b)	73.7 ± 3.2 (b)	4.1 ± 0.08 (d)	1.4 ± 0.3 (a)	0.5 ± 0.03 (c)	6.9 ± 0.6 (c)	5.5 ± 0.4 (c)
		5th	0.3 ± 0.03 (ab)	0.3 ± 0.01 (def)	242.7 ± 11 (c)	7.6 ± 0.04 (a)	62.4 ± 1.9 (a)	61.7 ± 9.5 (a)	13.1 ± 0.7 (d)	18.5 ± 0.7 (b)	21.6 ± 2.6 (b)	62.8 ± 4.5 (b)	4.9 ± 0.32 (c)	1 ± 0.2 (ab)	0.4 ± 0.07 (c)		0.3 ± 0.02 (e)
**M**	**Upper**	2nd			267.3 ± 24.7 (c)	7.7 ± 0.1 (a)	53 ± 2.2 (b)	45.2 ± 1.2 (ab)	19.7 ± 0.8 (b)	13.6 ± 0.8 (d)	4.8 ± 1.9 (b)	69.8 ± 2.5 (b)	4.3 ± 0.13 (dh)	1.5 ± 0.5 (a)	0.5 ± 0.01 (bc)	7.6 ± 0.25 (b)	4.5 ± 0.2 (d)
		5th	0.2 ± 0.04 (bc)	0.4 ± 0.02 (cd)	260.7 ± 10.1 (c)	7.4 ± 0.03 (a)	62.7 ± 1.1 (a)	39.6 ± 6.7 (ab)	16.5 ± 2.1 (c)	16 ± 2.1 (c)	21.9 ± 4 (b)	72.6 ± 3.5 (b)	5.3 ± 0.23 (c)	1.6 ± 0.1 (a)	0.4 ± 0.03 (c)		1 ± 0.3 (e)
	**Lower**	2nd			276 ± 42.8 (b)	7.6 ± 0.03 (a)	52.1 ± 1.4 (b)	37.5 ± 1 (ab)	20.3 ± 1.8 (b)	13.7 ± 1.8 (de)	3.9 ± 0.8 (b)	66.6 ± 2.2 (b)	4.1 ± 0.3 (h)	1.3 ± 0.1 (a)	0.5 ± 0.02 (bc)	7.7 ± 0.26 (b)	5.6 ± 0.4 (c)
		5th	0.2 ± 0.1 (bc)	0.4 ± 0.05 (cd)	238 ± 28.6 (c)	7.4 ± 0.04 (a)	62 ± 1.2 (a)	49.7 ± 8.1 (a)	15.2 ± 1.2 (c)	15.6 ± 1.2 (ce)	15.1 ± 6.4 (b)	63 ± 3 (b)	4.8 ± 0.14 (cd)	1.4 ± 0.1 (a)	0.4 ± 0.03 (c)		0.6 ± 0.2 (e)
**L_Bis**	**Upper**	2nd	0.4 ± 0.03 (a)	0.3 ± 0.02 (f)	187.7 ± 17 (c)	7.6 ± 0.1 (a)	52.9 ± 2.1 (b)	59.9 ± 1.6 (a)	13.7 ± 0.9 (d)	13.7 ± 0.9 (d)	6.7 ± 2.3 (b)	55.1 ± 3.4 (b)	5.6 ± 0.14 (b)	1.1 ± 0.2 (a)	0.5 ± 0.01 (bc)		1.3 ± 0.2 (c)
	**Lower**	2nd	0.3 ± 0.02 (bc)	0.4 ± 0.01 (cde)	218.3 ± 18 (c)	7.4 ± 0.04 (a)	57.2 ± 3.5 (a)	44.8 ± 1.6 (ab)	13.9 ± 0.9 (d)	15.7 ± 0.9 (c)	6.3 ± 1.5 (b)	60.1 ± 2.8 (b)	6.2 ± 0.1 (a)	1.5 ± 0.1 (a)	0.5 ± 0.03 (bc)		1.4 ± 0.5 (c)

Bold fonts: treatment names.

**Table 5 ijerph-17-03210-t005:** Estimated mass of Technosol and stock of nutrients per plot (0.64 m^2^). Initial stocks are calculated from concentrations in the parent materials and bulk densities when setting up the Technosols.

Layer/Total			Dry Weight (kg)	Corg (kg·plot^−1^)	CEC (cmol^+^.plot^−1^)	Ntotal (kg·plot^−1^)
L	Upper	Initial	23.2	5.3	0.9	0.3
		Year 5	29.9	5	1.5	0.8
	Lower	Initial	13	5.9	0.3	0.1
		Year 5	29.9	6.4	1.8	0.4
	**Total**	**Initial**	**36.2**	**11.2**	**1.2**	**0.3**
		**Year 5**	**59.7**	**11.4**	**3.2**	**0.8**
L-R	Upper	Initial	21.4	6	0.9	0.3
		Year 5	23.8	4.8	1.4	0.4
	Lower	Initial	13	6.5	0.4	0.2
		Year 5	23.6	5.7	1.5	0.4
	**Total**	**Initial**	**34.4**	**12.6**	**1.2**	**0.5**
		**Year 5**	**47.4**	**10.5**	**2.9**	**0.8**
M	Upper	Initial	18.1	5.6	0.6	0.2
		Year 5	19.3	5	1.2	0.3
	Lower	Initial	18	5.6	0.6	0.2
		Year 5	19.3	4.7	1.2	0.3
	**Total**	**Initial**	**36.2**	**11.2**	**1.2**	**0.3**
		**Year 5**	**38.6**	**9.8**	**2.5**	**0.6**
L_Bis	Upper	Initial	23.2	5.3	0.9	0.3
		Year 2	29.6	5.5	1.6	0.4
	Lower	Initial	13	5.9	0.3	0.1
		Year 2	19.2	4.2	1.1	0.3
	**Total**	**Initial**	**36.2**	**11.2**	**1.2**	**0.3**
		**Year 2**	**48.8**	**9.7**	**2.7**	**0.7**

**Table 6 ijerph-17-03210-t006:** Trace metal contents in Technosol and parent material. *NF U 44-551: AFNOR standard regarding potting media.

Year	Type	Layer	As	Cd	Cr	Cu	Pb	Hg	Zn	Ni	Se
			mg·kg^−1^ of DM								
NF U 44-551*				2	150	100	100	1	300	50	
**Parental material**	**Green waste compost**		5.9 ± 0.2 (ab)	0.5 ± 0.1 (a)	20.7 ± 1 (a)	36.8 ± 5.4 (b)	51.2 ± 4.4 (a)	0.3 ± 0.1 (a)	179.5 ± 6.9 (c)	9.4 ± 0.1 (a)	0.5 ± 0 (a)
	**Spent mushroom substrate**		4 ± 0 (b)	0.1 ± 0.08 (b)	5.9 ± 0.4 (c)	22.6 ± 4.7 (c)	3.1 ± 4.3 (b)	0.1 ± 0.1 (a)	26.7 ± 4.5 (d)	5 ± 0 (a)	0.5 ± 0 (a)
	**Crushed wood**		5.8 ± 3.1 (ab)	0.1 ± 0.08 (b)	13.3 ± 7.9 (b)	22.4 ± 20.5 (c)	9.1 ± 4.9 (b)	0.2 ± 0 (a)	34 ± 6.2 (d)	8.8 ± 6.1 (a)	0.5 ± 0 (a)
**Year 5**	**L**	Upper	5.2 ± 0 (ab)	0.5 ± 0.01 (a)	21.7 ± 0.9 (a)	42.5 ± 0.7 (ab)	60.3 ± 2 (a)	0.4 ± 0 (a)	200.5 ± 14.5 (a)	10.2 ± 0.4 (a)	0.7 ± 0 (a)
		Lower	4.9 ± 0.4 (ab)	0.5 ± 0.02 (a)	20.4 ± 1.3 (a)	43 ± 1.4 (ab)	59.5 ± 1.6 (a)	0.5 ± 0.02 (a)	247 ± 35.4 (a)	10.5 ± 0.5 (a)	0.5 ± 0 (a)
	**L-R**	Upper	4.5 ± 0.1 (ab)	0.5 ± 0.02 (a)	21.8 ± 0.5 (a)	53.3 ± 18.8 (ab)	58.6 ± 3.3 (a)	0.4 ± 0.05 (a)	190.7 ± 8.1 (a)	10 ± 0.5 (a)	0.5 ± 0.06 (a)
		Lower	4.5 ± 0.2 (ab)	0.5 ± 0.03 (a)	19.4 ± 0.8 (a)	50.7 ± 3.2 (ab)	53.7 ± 3.3 (a)	0.4 ± 0.06 (a)	209.7 ± 26.4 (b)	10.7 ± 0.5 (a)	0.5 ± 0 (a)
	**M**	Upper	4.5 ± 0.06 (ab)	0.5 ± 0.01 (a)	19 ± 0.6 (a)	43.3 ± 1.2 (ab)	59.1 ± 8 (a)	0.4 ± 0.01 (a)	187.7 ± 10.1 (b)	9.7 ± 0.8 (a)	0.5 ± 0 (a)
		Lower	5.2 ± 0.2 (ab)	0.5 ± 0.01 (a)	21 ± 1 (a)	45.7 ± 3.8 (ab)	68.5 ± 17.9 (a)	0.4 ± 0.07 (a)	207.3 ± 17.9 (b)	11 ± 0.5 (a)	0.6 ± 0.06 (a)
	**L_Bis**	Upper	5.8 ± 0.6 (ab)	0.5 ± 0.01 (a)	21.5 ± 1.9 (a)	40.7 ± 1.5 (ab)	80.4 ± 11.7 (a)	0.5 ± 0.09 (a)	210.3 ± 14.4 (b)	10.5 ± 0.3 (a)	0.5 ± 0 (a)
		Lower	4.7 ± 0.3 (ab)	0.5 ± 0.02 (a)	19.9 ± 0.6 (a)	40 ± 0 (ab)	61.7 ± 5.4 (a)	0.5 ± 0.01 (a)	197.3 ± 4.04 (b)	10 ± 0.3 (a)	0.8 ± 0.4 (a)

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
