# Peer review of "Potential of Technosols Created with Urban By-Products for Rooftop Edible Production"

_ijerph, 2020, doi:10.3390/ijerph17093210_

Round 1

Reviewer 1 Report

General comment:

This paper is of high interest due to its subject and its robust method that led to the acquisition of many data. Dear authors, please be careful about some terminological approximations, that deserved the paper. On the other hand, I would suggest to strengthen the scientific content of the work by highlighting what are the deep scientific questions behind the experiments. I also suggested a rephrasing of the title of the water which appears to me as irrelevant with its content.

Eventually – of course I let the editors on the decision - my suggestions would be to dig much more the results about the yields and the physico-chemical properties in this paper and to withdraw all the results about contamination. In my opinion, not only are these results a different topic, but their description in the actual state of the paper get the authors to go very fast on the agronomy of their system and this is really a pity to me considering the high value of this work. You may write a very interesting and relevant short communication about the contaminants in your system…

Specific comments:

L2: What about adding an S to “Technosol” as you studied many of them?

L22: Instead of « based only…” I would write « created only with urban wastes”

L42: the range of variation between 1.87 and 150 is amazingly huge; could you please comment that?

L52: Could you please highlight and emphasize what are the specific constraints and opportunities related to the food production on a roof compared to a ground production? We would expect more scientific questions in such an interesting paper than just comparing level of production…

L60: In my opinion, such a topic deserves the authors to spend few time to explain the differences between a soil (i.e. a living ecosystem) and its parent materials (e.g. urban wastes) and how human actions could contribute to such ecosystem creation

L63: I would say that urban wastes are “not sufficiently used resources” instead of “unused”; apart from that, be very clear on what is a waste (from a regulatory point of view) and what is a by-product (such as compost)

L66: Again, I would like the authors to explain why and how the use of urban wastes could raise some scientific questions (e.g. health risks, fertility)! You just started to mention that at L75, it is too late J

L81: ref (a) you wrote “Extensive mix extensive”, please correct

Table 1: such a review work has a potential huge interest!!! Unfortunately, its actual state lacks relevant information (e.g. the kind of food production, data about the yields), present some information under poor format (e.g. I would prefer to have the climatic conditions and the density of population instead of the location; emphasize the presence/absence of urban wastes) and show some irrelevant information (could you please explain the link between the creation of fertile substrate and the interest of irrigating… it could be of interest, but further explanations are required)

L93: My first reaction would be to say that the objectives of the paper are not in accordance with the title of the paper. Indeed, while reading “Designing fertile Technosol for edible production based on urban waste”, I would expect a paper dedicated to the design, that is to say the choice of the parent materials and their ratio… This is not what this paper is about. Therefore, I would suggest to change the title into something like “Potential of Technosols created with urban by-products for edible production”

L102: Please provide basic information about the climatic conditions, the height of the building, the density of population and if possible some data about the urban typology (e.g. housing, industrial, shopping)

L110: please correct “Error! Reference source not found”

L117: I would suggest to call them “constructed Technosols” or “created Technosol” to emphasize the positive approach in creating a soil; by the way, in this section, you don’t talk about creating a soil, but only present some parent materials

L118: I would suggest to use the term “parent materials” instead of “components”; be careful about the designation, some of your materials are not wastes but by-products!!! In addition, could you provide some basic physico-chemical data about them?

L136: This is the section where the design of the Technosols are describe. Please adapt the formulation of the title! Please separate the design of the soil from the experimental design by creating distinct sections

L169: You may consider merging Figure 1 and 2

L205: In my opinion, this experimental choice is strong and you should emphasize on it. Indeed, even if such a practice is of interest, I wonder how you could assess the sustainability of the soil’s fertility? You shall maybe consider mentioning the sustainability of the cropping design/maintenance to include this significant input of fresh parent materials

L219: could you explain which surface you considered for the calculation of the yields?

L270: please correct “Error! Reference source not found”

L268: I would expect much more details about the results that such general comments. You need to make the job and not letting the reader do everything on his/her own!!!! Apart from that, even if Table 3 is of real interest, I would recommend to try drawing few graphs for each species in order to show clearly the differences between treatments and over time…

L280: please correct “Error! Reference source not found”

L288: coming back to my comment about L205 => could we really consider that there is any ageing process while fresh materials were brought? Could you provide, in the discussion part more references about pedogenesis of Technosols?

L284, 289, 292, 294, 304, 310, 312, 314, 316, : please correct “Error! Reference source not found”

L291: such result about the absence of significant effect of the Technosol’s composition brings us back to my statement about inadequate title of the paper…

L300: Be careful!!! Shrinkage is a soil process mainly caused by clay minerals… I would say that in your case, shrinkage might not be the explanation… that may be either organic matter mineralization or settlement! Please stick to “decrease of the thickness”. You may also cite existing papers on the settlement of Technosols over time

Table 4: fonts are too small, very hard to read…

L339: Again, I would say that the description of the data is very poor compared to the interest of them!

L391: The addition of sections to this Discussion part would make it easier to read

L395: what is your method to get such a productivity value?

L480: “Developing rooftop farming requires to design specific growing systems fitted to the constraint of this environment.”: which are??? How did your work respond to that?

L482: “Rooftop cropping systems that use Technosols made of organic wastes allow managing easily soil contamination and fertility” How? Would you really say that it is EASY to use wastes and by-products?

L486: Please be more ambitious in the conclusion than out looking on other pollutants!!!

L492: Could you please, at least cite and describe the ongoing pedogenic processes? It would be of interest to question the potential contrast between pedogenesis and sustainability? I do believe that the authors have many things to say about it

Reviewer 2 Report

I think the paper is in general interesting, well written and international competitive. It is based on a long experimentation and this makes the reported results more robust. 

I think the author should better discuss:

  • the differences with professional producers (in some cases yields are really higher or lower)
  • what would happen in case of different crops
    • which crops could be better suited to technosols rooftop farming?
    • what woould happen with metal absorption (some crops exhibit a higher tendency in metals absorption)
  • an overall (even generic) cost analysis of the proposed rooftop farming with Technosol. I expect it will not be competitive compared to "traditional" farming, however some cost analysis (€ per kilogram of product) would be helpful for the reader. 

Table 2 rendering should be improved

Figure 1 (on the right) is not completely clear: why not inserting google map image?

https://www.google.com/maps/place/48%C2%B050'24.4%22N+2%C2%B020'54.5%22E/@48.8400086,2.3483461,76m/data=!3m1!1e3!4m5!3m4!1s0x0:0x0!8m2!3d48.8401111!4d2.3484722
